# COVID-19 Disease in Pediatric Solid Organ Transplantation from Alpha to Omicron: A High Monocyte Count in the Preceding Three Months Portends a Risk for Severe Disease

**DOI:** 10.3390/v15071559

**Published:** 2023-07-16

**Authors:** Yasmina Sirgi, Maja Stanojevic, Jaeil Ahn, Nada Yazigi, Stuart Kaufman, Khalid Khan, Bernadette Vitola, Cal Matsumoto, Alexander Kroemer, Thomas Fishbein, Udeme D. Ekong

**Affiliations:** 1Department of Surgery, Georgetown University Medical School, Washington, DC 20007, USA; ysirgi@gmail.com; 2Department of Pediatrics, MedStar Georgetown University Hospital, Washington, DC 20007, USA; maja_stanojevic@outlook.com; 3Department of Biostatistics, Bioinformatics, and Biomathematics, Georgetown University, Washington, DC 20007, USA; ja1030@georgetown.edu; 4MedStar Georgetown Transplant Institute, MedStar Georgetown University Hospital, Washington, DC 20007, USA; nada.a.yazigi@gunet.georgetown.edu (N.Y.); stuart.s.kaufman@gunet.georgetown.edu (S.K.); khalid.m.khan@gunet.georgetown.edu (K.K.); bernadette.vitola@medstar.net (B.V.); cal.s.matsumoto@gunet.georgetown.edu (C.M.); alexander.kroemer@gunet.georgetown.edu (A.K.); thomas.m.fishbein@gunet.georgetown.edu (T.F.)

**Keywords:** immunocompromised children, liver transplant, viral infection, intestine transplant, lymphocyte count, innate immunity, inflammation

## Abstract

Importance: Planning for future resurgences in SARS-CoV-2 infection is necessary for providers who care for immunocompromised patients. Objective: to determine factors associated with COVID-19 disease severity in immunosuppressed children. Design: a case series of children with solid organ transplants diagnosed with SARS-CoV-2 infection between 15 March 2020 and 31 March 2023. Setting: a single pediatric transplant center. Participants: all children with a composite transplant (liver, pancreas, intestine), isolated intestine transplant (IT), isolated liver transplant LT), or simultaneous liver kidney transplant (SLK) with a positive PCR for SARS-CoV-2. Exposure: SARS-CoV-2 infection. Main outcome and measures: We hypothesized that children on the most immunosuppression, defined by the number of immunosuppressive medications and usage of steroids, would have the most severe disease course and that differential white blood cell count in the months preceding infection would be associated with likelihood of having severe disease. The hypothesis being tested was formulated during data collection. The primary study outcome measurement was disease severity defined using WHO criteria. Results: 77 children (50 LT, 24 intestine, 3 SLK) were infected with SARS-CoV-2, 57.4 months from transplant (IQR 19.7–87.2). 17% were ≤1 year post transplant at infection. 55% were male, 58% were symptomatic and ~29% had severe disease. A high absolute lymphocyte count at diagnosis decreased the odds of having severe COVID-19 disease (OR 0.29; CI 0.11–0.60; *p* = 0.004). Conversely, patients with a high absolute monocyte count in the three months preceding infection had increased odds of having severe disease (OR 30.49; CI 1.68–1027.77; *p* = 0.033). Steroid use, higher tacrolimus level, and number of immunosuppressive medications at infection did not increase the odds of having severe disease. Conclusions and relevance: The significance of a high monocyte count as predictor of severe disease potentially confirms the importance of monocytic inflammasome-driven inflammation in COVID-19 pathogenesis. Our data do not support reducing immunosuppression in the setting of infection. Our observations may have important ramifications in resource management as vaccine- and infection-induced immunity wanes.

## 1. Introduction

Over the course of 2021 and 2022, COVID-19 disease following SARS-CoV-2 infection and its variants became ubiquitous in human communities across the globe, including in children with solid-organ transplants (SOT). Studies in the adult population show increased hospitalization and mortality rates in immunocompromised patients, including those with SOT [1,2]. However, data on COVID-19 disease in pediatric patients with liver, pancreas, and intestine SOT is scarce. As immunosuppressed children have different responses to viral infection, we sought to determine the factors associated with disease severity in a cohort of children with a composite transplant (liver, pancreas, intestine), isolated intestine transplant (IT), isolated liver transplant (LT), or simultaneous liver kidney transplant (SLK). We hypothesized that children on the most immunosuppression, defined by the number of immunosuppressive medications and usage of steroids, would have the most severe disease course and that differential white blood cell count in the months preceding infection would be associated with likelihood of having severe disease.

## 2. Methods

### 2.1. Patients

This is a case series that describes children with solid organ transplants diagnosed with SARS-CoV-2 infection between 15 March 2020 and 31 March 2023, followed at a single pediatric transplant center. SARS-CoV-2 infection was diagnosed with a positive SARS-CoV-2 nucleic acid test (NAT) on a nasopharyngeal sample or bronchoalveolar sample, or a positive rapid antigen test at home or in the hospital. Indications for testing included symptoms, known exposure to COVID-19, or hospital policy prior to procedures, specifically upper endoscopy and colonoscopy, ileoscopy, or liver biopsy. Data extracted from medical records included demographic information, type of transplant, date of transplant, date of infection, number and type of immunosuppressant medication(s) at time of infection (tacrolimus, sirolimus, mycophenolate, steroids, and infliximab), tacrolimus trough at infection, steroid dose at infection (high-dose steroid was defined as ≥0.5 mg/kg/day; low-dose steroid was defined as <0.5 mg/kg/day), absolute lymphocyte count and absolute monocyte count at infection and three months preceding infection, natural killer (NK) cell frequency and number preceding infection, history of biopsy-proven acute rejection in the six to twelve months preceding infection, and presence of comorbidities at infection. Comorbidities recorded included obesity, hypertension, hyperlipidemia, diabetes, and chronic lung disease. Obesity was defined as a body mass index (BMI) at or greater than the 95th percentile for children of the same age and sex according to the CDC definition: https://www.cdc.gov/obesity/childhood/defining.html (accessed on 15 March 2020). Chronic lung disease was defined as a history of asthma or reactive airway disease. Hypertension was identified by the use of anti-hypertensive drugs, diabetes identified by the use of insulin, and hyperlipidemia identified by the use of fenofibrate. Chronic kidney disease was identified from the patient problem list in the electronic medical record. COVID-19 disease severity was stratified using the WHO criteria as follows: 1: no limitation of activity, 2: limitation of activity, 3: hospitalized, no oxygen therapy, 4: oxygen by mask or nasal prongs, 5: non-invasive ventilation or high flow oxygen, 6: intubation and mechanical ventilation, 7: ventilation and additional organ support. Severe disease was defined as WHO score ≥ 3. With regards to the prevailing variant in the community at time of infection, patients were categorized by date of infection as follows: March 2020–May 2021 other; June 2021–December 2021 delta variant; January 2022–March 2023 omicron variant. Acute rejection of the liver graft, intestinal graft, or kidney graft was defined as previously described [3,4,5,6].

### 2.2. Statistics

Demographical and clinical variables were descriptively summarized for the severe and mild disease groups. Paired samples *t*-tests or Pearson’s chi-squared test were used to compare continuous or categorical variables between severe and mild disease groups, respectively. Multivariable logistic regression including the variables (*p* < 0.2 or *p* < 0.05) in the previous bivariate comparisons was performed, and odds ratios (OR) and their 95% confidence intervals (CI) were summarized. For the purposes of analysis, the isolated intestine transplant group and composite transplant group were collapsed into a single group named intestine transplant. A two-sided significance level of 0.05 was used for statistical significance. Pearson’s correlations among the observed levels of NK cell number and frequency, CRP, LDH, ferritin, absolute monocyte and lymphocyte count were computed. This study was approved by Georgetown University IRB (Study Number: 2017-0365).

## 3. Results

### 3.1. Demographics and Clinical Course

Over the timeframe of the study, 77 patients tested positive for SARS-CoV-2, of which 49 patients received an isolated liver transplant, eight patients an isolated intestine transplant, 17 patients a composite (liver/pancreas/intestine) transplant, and three patients a simultaneous liver kidney transplant. Four patients had more than one documented infection, but only one infection was included; 55 patients had mild COVID-19 disease and 22 patients had severe disease. Of the 77 patients, 33 were asymptomatic and tested positive either at pre-procedure testing or following exposure to COVID-19. Of the 44 symptomatic patients, 22 required hospitalization; two patients were admitted to PICU for septic shock with one patient requiring endotracheal intubation and ventilation; three other patients were admitted to PICU for respiratory support. All patients completely recovered. One patient who was initially asymptomatic at the time of diagnosis subsequently required hospitalization for bleeding at the ostomy site on day 6 of infection and oxygen supplementation on day 16 of infection. Complete details of all the patients are summarized in Table 1.

### 3.2. High Absolute Monocyte Count in the Three Months Preceding Infection and High Tacrolimus Level Are Associated with Severe COVID-19 Disease in Univariate Analysis

In the univariate analysis, a high absolute monocyte count in the three months preceding infection, a high tacrolimus level at diagnosis of infection, a shorter duration from transplant at infection, a history of chronic lung disease, and a low absolute lymphocyte count at diagnosis of infection were all significantly associated with severe COVID-19 disease. Of the 77 patients with SARS-CoV-2 infection, 13 were less than one year post transplant at time of infection.

A history of hypertension at the time of diagnosis trended towards an association with severe COVID-19 disease, but did not achieve statistical significance. African Americans showed a trend towards severe disease (Table 2). A higher mean NK cell frequency before infection was seen in subjects with severe disease (14.1 ± 7.6% vs. 7.8 ± 7.2%; *p* = 0.08), and while mean ferritin levels were higher at diagnosis in subjects with severe disease (320.5 ± 453.7 ng/L vs. 175.3 ± 427.9 ng/L; *p* = 0.4), this association did not achieve statistical significance. Neither age, sex, type of solid organ transplant, number of immunosuppressant medications, nor steroid use were associated with an increased risk of severe COVID-19 disease. Similarly, COVID-19 disease did not appear to increase in severity with changes in the predominant viral variant in the community. Other comorbidities (aside from hypertension) reported to be risk factors for disease severity in adults were not associated with disease severity in our pediatric solid organ transplant cohort in the univariate analysis.

### 3.3. Concomitant Immunosuppression, Steroid Dose, and Protection against Severe Disease

To dig more deeply into the role concomitant immunosuppressive medications such as m-TOR inhibitors, anti-metabolites could play, either as protection against severe disease or in driving disease severity, we compared disease severity in those receiving m-TOR inhibitors/antimetabolites vs. those not receiving these medications. Given the recommendation of dexamethasone for treatment of moderate to severe COVID-19 disease in the National Institutes of Health (NIH) coronavirus disease (COVID-19) treatment guidelines at https://www.covid19treatmentguidelines.nih.gov (Updated 26 September 2022. accessed on 18 March 2023) [7], we investigated for evidence of a protective effect from high dose steroids using steroid dose per kilogram body weight of patients at time of infection. Thirty-one patients were on low dose steroids, two patients were on high dose steroids, and the remaining 44 patients were not receiving steroids. Steroid dose was not associated with a protective effect against severe disease. Seventeen patients were on sirolimus at time of infection; sirolimus was also not associated with a protective effect against or increased risk for severe disease. Only one patient was on infliximab at time of infection (Table 2 and Table 3).

### 3.4. Acute Rejection and COVID-19 Disease

To more broadly look for a potential relationship between acute rejection and COVID-19 disease in general, we examined the frequency of acute rejection episodes in the 6–12 months preceding infection, in addition to the severity of acute rejection episodes. Fifteen patients had biopsy-proven acute rejection (BPAR) a median (IQR) of 47.5 (26.3–57.9) months preceding SARS-CoV-2 infection. Three of these patients had BPAR ≤12 months before SARS-CoV-2 infection (Table 1). The BPAR episodes occurring ≤12 months before infection were mild in severity. Acute rejection in the months preceding infection was not associated with a risk of developing severe disease (Table 2).

### 3.5. High Lymphocyte Count at Diagnosis Decreased, and High Monocyte Count Preceding Infection Increased, the Odds of Having Severe COVID-19 Disease in Multivariable Analysis

A high absolute lymphocyte count at diagnosis decreased the odds of having severe COVID-19 disease (OR 0.29; CI 0.11–0.60; *p* = 0.004); Conversely, patients with a high absolute monocyte count in the three months preceding infection had increased odds of having severe disease (OR 30.49; CI 1.68–1027.77; *p* = 0.033); however, a longer duration from transplant at infection, a higher tacrolimus level at infection, and chronic lung disease did not significantly decrease or increase the odds of having severe disease in the multivariable analysis (Figure 1).

### 3.6. Evolution of Acute Phase Reactants

Thirty-one patients had ferritin, 15 had LDH, and 10 patients had CRP measured over the course of infection up to 60 days following diagnosis of infection. CRP appeared to normalize by 30 days following infection. One patient had elevated ferritin as far out as 60 days following infection, and a second patient still had elevated LDH as far out as 60 days following infection (Figure 2).

We computed Pearson’s correlations (Appendix A). NK cell frequency before infection was negatively correlated to ferritin level at time of infection (r = −0.95; *p* < 0.01), and NK cell count before infection was negatively correlated to LDH level at time of infection (r= −0.89; *p* < 0.01); however, the small number who had these acute phase reactants and NK cell measurements (20 subjects) precludes drawing any major conclusions.

## 4. Discussion

SOT patients are at the highest risk of infection, including respiratory viral infections, within the first post-transplant year [8] and therefore, as a transplant center that only paused our living donor and not our deceased donor program during the pandemic, our observation that most patients were over four years from transplant at time of infection was surprising. Indeed, only 13 of 77 patients were less than one year from transplant when they were infected. It is plausible that parents of newly transplanted children were hyper-alert given the novelty of both the transplant procedure to their nuclear family unit and COVID-19 disease and the ramifications of infection in a heavily immunosuppressed transplant recipient in general. Correlating with our observation, Bansal et al. reported a median duration from transplant to infection of 3.4 years in their cohort of pediatric solid organ transplant recipients [9]. They had no deaths in their cohort and all patients fully recovered following infection.

In our cohort, 42% of patients who were infected were asymptomatic. Our observation is different from the report by Varnell et al., who, using the IROC database, found that of 281 pediatric kidney transplant recipients tested, 24 (8.5%) were positive, 15 (63%) were symptomatic and 37% were asymptomatic. The difference in the observations from the two studies is probably reflective of the longer duration of our study, March 2020 through March 2023, compared to the Varnell et al. report, which ran from April 2020 through September 2020 [10]. Moreover, the patient cohort in both studies was different. The variants prevalent in the community during the period of our study were also different. The median duration after transplant at infection in their cohort was shorter, 2.9 years, compared to our cohort. These may all have influenced the differences observed in clinical presentation following infection in the two cohorts. A meta-analysis of COVID-19 disease in children by Cui X et al. [11] suggests the frequency of asymptomatic infection to be 20% (95% CI 14–26%). Notably, the meta-analysis includes only published studies from December 2019 through April 2020, which could explain the lower frequency compared to our cohort; moreover, testing was conducted more frequently in our cohort compared to what would pertain in otherwise healthy children in the general population.

As in reports from adult cohorts, lymphopenia at diagnosis was associated with a more severe clinical course [12]. What remains unclear is whether lymphopenia is a consequence of SARS-CoV-2 infection or a consequence of calcineurin inhibitors and steroids. We tried to investigate this by examining absolute lymphocyte counts in the three months preceding infection and surprisingly observed a lower absolute lymphocyte count in those with mild compared to severe disease (Table 2). Viral-induced apoptosis has been suggested to explain lymphopenia in COVID-19 disease [13]. Monocytopenia has been described in COVID-19-positive patients, with some studies reporting lower monocyte counts in ICU patients [14,15]. Others have reported a higher percentage of CD14^+^CD16^+^ inflammatory monocytes in patients with COVID-19 disease compared to normal controls [16]. We did not observe significant differences in monocyte count at diagnosis of SARS-CoV-2 infection in our cohort; however, quite interestingly, a high monocyte count in the three months preceding infection increased the odds of developing severe disease 30.4-fold in our cohort. This observation is notable as monocytes are thought to play a significant role in sustaining the hyperinflammatory response in SARS-CoV-2 infection. Indeed, the significance of a high monocyte count as a predictor of severe disease potentially confirms the importance of monocytic inflammasome-driven inflammation in COVID-19 pathogenesis, as shown by research from us and others [17,18,19]. It is important to note that the hyperinflammatory response associated with monocytes has also been reported in other infections, specifically HIV infection, sepsis [20], and tick-borne hemorrhagic fever [21], and in patients with acutely decompensated cirrhosis who develop acute-on-chronic liver failure [22]. The negative correlation of NK cell frequency and number before infection with ferritin and LDH levels at time of infection, respectively, is interesting. While subjects with severe disease trended towards a higher NK cell frequency before infection, the limited number (20 subjects) who had this measurement taken precluded its addition to the multivariable analysis. Importantly, we do not know if the higher NK cell frequency was maintained at the time of infection. Contraction of NK cells has been reported in adults with fatal disease [23]. NK cells are known to exert primary control during viral infection [24] and the loss of NK cells in adults with COVID-19 pneumonia and severe disease is postulated to be a consequence of the recruitment of NK cells into infected tissues [25,26].

Fifty-one patients (66%) had at least one comorbidity, including hypertension, chronic lung disease, diabetes, chronic kidney disease, hyperlipidemia, and obesity. We examined the potential impact of specific comorbidities on the risk of severe disease. In our cohort, only chronic lung disease was significantly associated with severe disease in our univariate analysis, though it did not reach significance in our multivariate analysis. This observation is different from published observations in adults with SARS-CoV-2 infection [27]. It is possible that our small sample size (only 8 of 77 patients had obesity) precluded observation of any significant association, as a meta-analysis of 42 studies that included 275,661 children showed a higher risk of severe COVID-19 and COVID-19-related mortality among children with comorbidities comparing to healthy children. The authors of the meta-analysis examined the risk of obesity on COVID-19 severity in relation to children without comorbidities and found a relative risk ratio of 2.87 (95% CI 1.16–7.07; *p* = 0.17), and concluded that childhood obesity likely increases the risk of severe COVID-19; however, more case-controlled, well-defined studies were needed to examine the effects that other childhood comorbidities have on risk of severe COVID-19 disease [28]. Given our small numbers, we are unable to comment on the role our immunosuppressive agents, specifically calcineurin inhibitors and systemic steroids, may have had, if any, in reducing baseline levels of cytokines such as IL-6 thought to be elevated in obesity, and contributory to increased susceptibility to severe COVID-19 disease. Moreover, our definitions of high and low steroid exposure may not have been sufficient to identify a positive or negative steroid effect, since variables such as time after transplant, hypertension, number of immunosuppressive agents, and steroid use might be co-associated.

## 5. Study Strengths

Our study has several important strengths. From a pediatric perspective, our sample size of 77 patients is significant and is one of the few published studies with outcomes reported in many children with composite and isolated intestine transplants. Furthermore, our study is the first to show that a low lymphocyte count and a high monocyte count in the three months preceding diagnosis increases the odds for severe COVID-19 disease. This observation has implications for resource management in the event of a resurgence in SARS-CoV-2 infection as vaccine- and as infection-induced immunity wanes. Last but not least, our study covers the entire period of the pandemic with cases representing the spectrum of prevailing variant(s) in the community, thereby providing information on expected outcomes.

## 6. Study Limitations

Our study has several potential limitations. First, our criteria for severe disease were wide and included WHO score 3 patients, i.e., admitted to hospital but no oxygen, ventilatory or organ support requirements. Second, the decision to admit to hospital could be a result of admission selection bias in favor of children with solid organ transplants. Third, asymptomatic infection may have occurred in recipients that was not captured if they did not have a reason for testing, such as a planned clinical procedure or following a known exposure. Such a broader representation would have increased the accuracy of risk computation in this patient population.

## 7. Conclusions

We report that lymphopenia increases the odds for severe COVID-19 disease 0.29-fold in children with solid organ transplants. In particular, a high monocyte count in the three months preceding diagnosis increases the odds for severe COVID-19 disease 30.4-fold in children with solid organ transplants. These observations may have important ramifications in resource management in the event of a subsequent resurgence in SARS-CoV-2 infection as vaccine- and infection-induced immunity wanes.

## Figures and Tables

**Figure 1 viruses-15-01559-f001:**
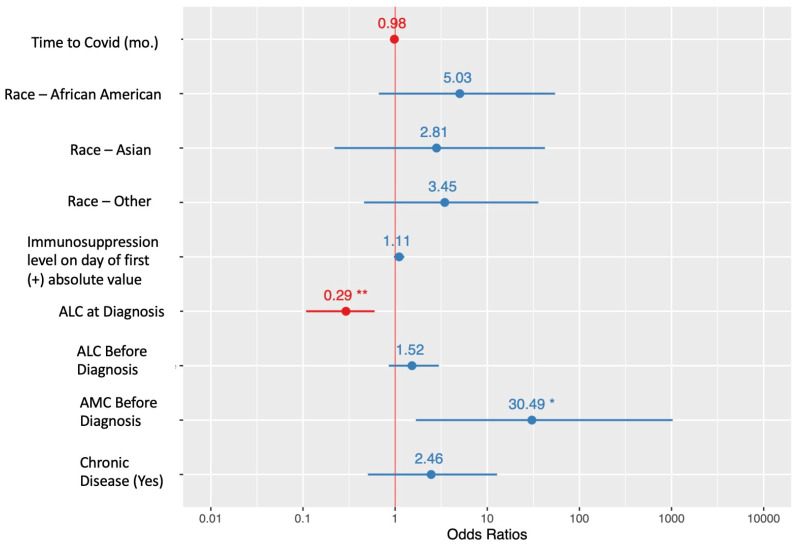
Forest plot of odds ratios (OR) from multivariable logistic regression. OR > 1 (blue) indicates a risk factor for severe symptoms and OR < 1 (red) indicates a protective factor. * *p* < 0.05 ** *p* < 0.01.

**Figure 2 viruses-15-01559-f002:**
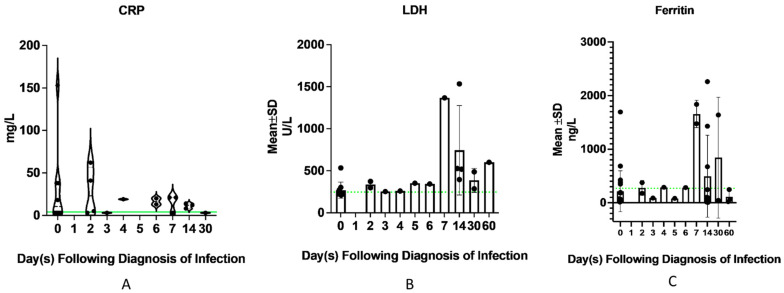
Trend of acute phase reactants over the course of infection. (**A**) Ten patients had C-reactive protein (CRP) measurements performed at time of diagnosis of infection and between 0 to 30 days following infection. CRP appeared to have normalized by day 30 following infection. The green dotted line denotes the upper limit of normal. (**B**) Fifteen patients had lactate dehydrogenase (LDH) measured at time of diagnosis of infection and between 0 to 60 days following infection. One patient continued to have an elevated LDH level even at 60 days following infection. The green dotted line denotes the upper limit of normal. (**C**) Thirty-one patients had ferritin measured at time of diagnosis of infection and between 0 to 60 days following infection. One patient still had elevated ferritin levels as far as 30 days following infection. Of those who had ferritin measured at 60 days following infection, the levels had normalized. The green dotted line denotes the upper limit of normal.

**Table 1 viruses-15-01559-t001:** Demographics.

Variables	*n* (%)
Transplant type	
▪ isolated liver	50 (65)
▪ intestine (composite + isolated intestine)	24 (31)
▪ simultaneous liver kidney	3 (4)
Symptoms	
▪ yes	44 (58)
▪ no	33 (42)
Gender	
▪ male	42 (55)
▪ female	35 (45)
Duration from transplant at infection (months)	
• median (IQR)	57.4 (19.7–87.2)
≤1 year from transplant at infection	13 (17)
>1 year from transplant at infection	64 (83)
Not hospitalized	55 (71)
Hospitalized	22 (29)
▪ PICU	5 (23)
▪ Floor	17 (77)
Interval between BPAR in preceding 12 months and infection (months)	
• median (IQR)	1.67 (1.085–3.92)

Key: BPAR: biopsy proven acute rejection. PICU: pediatric intensive care unit. IQR: interquartile range.

**Table 2 viruses-15-01559-t002:** Univariate analysis of variables.

	* Mild Disease	Severe Disease	*p*-Value
	(*n* = 55)	(*n* = 22)	
Age at diagnosis (years)			
Mean (SD)	7.99 ± 4.83	6.91 ± 4.57	0.37
Sex			
Male	29 (53)	13 (59)	0.8
Female	26 (47)	9 (41)	
Race			
African American	14 (25)	11 (50)	0.06
White	20 (36)	2 (9)	
Asian	7 (13)	4 (18)	
Other	14 (25)	5 (23)	
Ethnicity			
Non-Hispanic	42 (76)	18 (82)	0.82
Hispanic	13 (24)	4 (18)	
Duration from transplant at infection (months)			
Mean (SD)	66.40 ± 45.96	42.09 ± 40.02	0.03
Tacrolimus level (ng/dL)			
Mean (SD)	6.0 ± 5.3	8.9 ± 7.2	0.05
On predniso(lo)ne at time of infection			
yes	22 (41)	14 (64)	0.11
no	32 (59)	8 (36)	
Steroid dose			
low	21 (38)	13 (59)	0.11
high	1 (2)	1 (5)	
not on steroids	33 (60)	8 (36)	
Chronic lung disease			
no	42 (76)	11 (50)	0.04
yes	13 (24)	11 (50)	
Hypertension			
yes	16 (29)	12 (55)	0.06
no	39 (71)	10 (45)	
Hyperlipidemia			
yes	6 (11)	3 (14)	1
no	49 (89)	19 (86)	
Chronic kidney disease			
yes	16 (29)	7 (32)	1
no	39 (71)	15 (68)	
Obesity			
yes	5 (9)	3 (14)	0.8
no	50 (91)	18 (86)	
History of acute rejection			
no	45 (82)	17 (77)	0.89
yes	10 (18)	5 (23)	
Number of immunosuppression medications at infection			
1	20 (37)	4 (18)	
2	22 (41)	10 (45)	0.21
3	12 (22)	8 (36)	
Type of transplant			
Isolated liver	37 (67)	13 (59)	0.31
Simultaneous liver kidney	3 (5)	0 (0)	
Intestine (composite + isolated intestine)	15 (27)	9 (41)	
Absolute lymphocyte count at infection (K cells/μL)			
mean ± SD	2.77 ± 1.43	1.79 ± 1.21	0.006
Absolute monocyte count at infection (K cells/μL)			
mean ± SD	0.55 ± 0.29	0.66 ± 0.48	0.23
Absolute lymphocyte count before infection (K cells/μL)			
mean ± SD	2.68 ± 1.19	3.21 ± 2.31	0.18
Absolute monocyte count before infection (K cells/μL)			
mean ± SD	0.49 ± 0.20	0.75 ± 0.73	0.01
On mycophenolate at time of infection			
yes	13 (24)	6 (27)	0.96
no	42 (76)	16 (73)	
On sirolimus at time of infection			
yes	10 (18)	5 (23)	0.89
no	45 (82)	17 (77)	
SARS-CoV-2 prevailing variant			
Omicron	25 (45)	10 (45)	0.91
Other	22 (40)	8 (36)	
Delta	8 (15)	4 (18)	

*p*-value was obtained using paired samples *t*-test or Pearson’s chi-squared test. * Asymptomatic patients included in the dichotomous comparisons of mild vs. severe disease.

**Table 3 viruses-15-01559-t003:** Immunosuppression by graft type.

Immunosuppression	Transplant Type			
	Isolated Liver	Isolated Small Bowel	Composite	Simultaneous Liver Kidney
Tacrolimus	49	7	17	3
Sirolimus	5	3	6	3
Mycophenolate	17	1	1	0
Predniso(lo)ne				
Low dose	7	7	15	2
High dose	2	0	0	0
Infliximab	0	0	1	0

## Data Availability

Data available upon request.

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
