# Peer review of "COVID-19 Disease in Pediatric Solid Organ Transplantation from Alpha to Omicron: A High Monocyte Count in the Preceding Three Months Portends a Risk for Severe Disease"

_viruses, 2023, doi:10.3390/v15071559_

Round 1
Reviewer 1 Report
They report that lymphopenia increases the odds for severe COVID-19 disease 0.27- fold in children with solid organ transplants. In particular, a high monocyte count in the three-months preceding diagnosis increases the odds for severe COVID-19 disease 22.6- fold in children with solid organ transplants. These observations may have important ramifications in esource management in the event of a subsequent resurgence of SARS- CoV-2 infection as vaccine and infection-induced immunity wanes.
Suggestions?
1. why did you check the monocyte count in the three-months preceding diagnosis but not four-months or five-months preceding diagnosis?
2. Monocytes are thought to play a significant role in sustaining the hyperinflammatory response in SARS-CoV-2 infection, is this phenomenon specific for SARS- CoV-2 infection?
Minor editing of English language required.
Reviewer 2 Report
It is an interesting study and it is a continual of the previous study published by the same group (PMID: 35758426) about the immune cells changes before/ after COVID infection in pediatric solid organ transplants and how these changes affect the disease course.
Major comments:
1) It is important not only show the immune cell (either monocytes/ or lymphocytes( before or after infection, but the authors should assess the cytokine storms especially that associated with rejection in the patients' plasma.
2) Also one important point is the NK cell levels, what is the level of these cells in those patients before the infection? could this affect the disease course?
3) Figure 2: can the authors correlate to CRP, LDH, ferritin to monocyte, lymphocytes at the same time point. Also the level of immunosuppressive in the same sample
4) COVID-19 load should be provided and correlated to all parameters.
5) Patients' ethnicity should be provided,
Language is fine
Round 2
Reviewer 2 Report
No further comments
Language is fine